# Choose Before You Label: Efficient Node and Data Selection in Distributed Learning

## Abstract

We consider one of the most relevant problems of distributed learning, i.e., the selection of the learning nodes to include in the training process as well as the selection of the samples from each of the learning nodes' local datasets, so as to make learning sustainable. Traditional approaches rely on pursuing a balanced label distribution, which requires label statistics from *all* datasets, including those not selected for learning. This may be costly and may raise privacy concerns. To cope with this issue, we aim at selecting few and small datasets. To this end, we propose a new metric, called *loneliness*, which is defined on *unlabelled* training samples. First, through both a theoretical and an experimental analysis, we show that loneliness is strongly linked with learning performance (i.e., test accuracy). Then, we propose a new node- and data-selection procedure, called Goldilocks, that uses loneliness to make its decisions. Our performance evaluation, including three state-of-the-art datasets and both centralized and federated learning, demonstrates that Goldilocks outperforms approaches based upon a balanced label distribution by providing over 70% accuracy improvement, in spite of using information that is both less sensitive privacy-wise and less onerous to obtain.

## 1 Introduction

Node and data selection are two of the foremost issues in distributed machine learning, e.g., federated learning (FL), requiring to strike a difficult balance between having enough data to properly learn, and limiting the resource consumption and cost of the overall learning process. Such issues are especially relevant in scenarios where dependable, sustainable learning is required, e.g., for safety-critical applications at the network edge. In these scenarios, it is important to guarantee a given level of learning quality (e.g., test accuracy) while being able to involve in the training process heterogeneous nodes with different memory and computing capabilities. It follows that selecting only few, small local datasets becomes crucial, since including additional nodes in the model training—or using more data from the already-included nodes—may increase resource consumption whilst resulting in longer training times, poorer learning performance, or both. Avoiding these issues—intuitively, identifying the nodes *and* data that are worth the additional expense and effort—is the overarching goal of all node and data selection strategies.

The problem is compounded by the fact that evaluating the quality of the local dataset of a potential learning node is a very hard problem. Consider for simplicity a classification problem. The vast majority of current approaches (Wu & Wang, 2022; Li et al., 2021; 2022) use as discriminant the *labels* of data. Specifically, they associate a higher quality to balanced datasets in which all classes are equally represented, and they combine local datasets from different nodes in such a way that the resulting global training data is as close to balanced (often also referred to i.i.d.) as possible. However, sharing label information impinges the learning nodes' privacy, which might be problematic in the case of highly sensitive data. Additionally, in practical scenarios, labels can be expensive to obtain. While this is unavoidable for datasets that will eventually be used for training, going through the expense and hassle of labeling a dataset only to find it unsuited for the current learning task would again result in a waste of time and resources.

In this paper, we address the above issues by proposing a new metric, called *loneliness*, that can be effectively used to evaluate the quality of *unlabelled* datasets, hence, how suited they are to the learning task being carried out, while preserving data privacy. Loneliness is linked to how far away

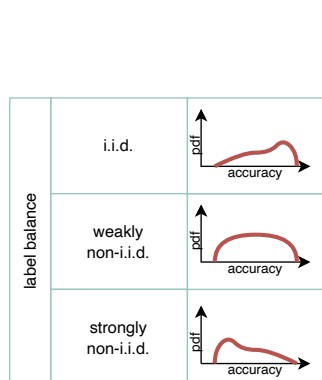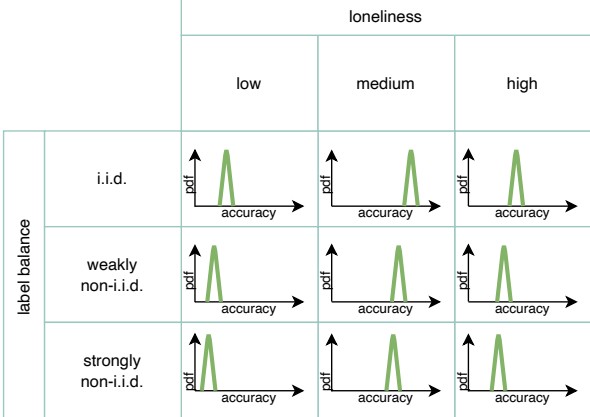

Figure 1: A qualitative depiction of how using loneliness improves testing accuracy. The figure compares the information accounted for by state-of-the-art approaches based on label balance (left) and Goldilocks (right); each plot illustrates pictorially the empirical distribution of test accuracy achievable by training sets with a given level of loneliness and label balance. Accounting for label distribution only provides partial information on the learning outcome. On the contrary, loneliness has a much stronger correlation with accuracy; therefore, by leveraging on it, Goldilocks is able to make better decisions, hence, achieving higher accuracy.

samples within a dataset are from each other, or, intuitively, to how many stand apart from the other samples of the dataset, with higher levels of loneliness corresponding to larger numbers of such samples.

Through a set of experiments, we show that (i) loneliness exhibits a stronger correlation than label balance with the testing performance of a learning task, and that (ii) the best performance is associated with intermediate loneliness values. As a theoretical explanation of this second finding, we provide a probably-approximately correct (PAC) Bayes bound, based on adaptive subspace compression (Lotfi et al., 2022). Through this bound, we illustrate that while the training error increases monotonically with the loneliness, the number of compressed bits to represent the weights of the deep neural network (DNN) (which we use as complexity term in the PAC-Bayes bound) decreases with the loneliness.

Inspired by our theoretical and experimental findings, we propose a node- and data-selection procedure, called *Goldilocks*, that exploits loneliness to make the inter-related decisions of (i) which nodes to include in the process, and (ii) which of their local data to use for the model training. By making such decisions jointly, Goldilocks is able to explore a wide range of high-quality trade-offs between the effectiveness of the learning process and the resources this necessitates. Furthermore, Goldilocks does not require the use of label information, but it can leverage it if available. Finally, and very remarkably, it can identify the most promising datasets for which it is worth it to obtain labels.

A pictorial sketch of how our approach and the Goldilocks procedure improve over the state of the art is provided in Fig. 1. The traditional approach, represented on the left, is geared towards scenarios where we must select tens or hundreds of learning nodes and makes decisions based on how balanced labels are. While it is true that better-balanced datasets yield better performance, there is still a significant variability within the accuracy yielded by similarly-balanced datasets. Our approach, represented on the right, leverages the loneliness metric (along with label information, if available), resulting in a much more accurate knowledge of which datasets yield the best test accuracy, especially when datasets are small and only a small number thereof can be selected.

We evaluate the performance of Goldilocks in both centralized and federated scenarios, using state-of-the-art DNN models and datasets, and find it to consistently outperform approaches only considering label information. Importantly, the performance metrics we consider go beyond mere classification accuracy, and include the number of learning nodes to involve in the learning process and the quantity of data therein to exploit.

In summary, our main contributions can be summarized as follows:

- We propose a new metric, called *loneliness*, estimating the suitability of an *unlabeled* dataset (and, hence, of the learning node owning the data) for a given learning task;

- We perform a set of experiments, demonstrating that a strong link exists between loneliness and learning performance;

- We provide a theoretical explanation for the effectiveness of loneliness, based on model compression and a PAC-Bayes bound;

- We leverage loneliness and the insights provided by the theoretical analysis to design a procedure, called Goldilocks, that makes high-quality node *and* data selection decisions;

- We evaluate the performance of Goldilocks under both centralized and FL tasks using multiple datasets and neural networks, demonstrating that it consistently finds the best trade-offs between the resources needed for training and the resulting test accuracy. Notably, Goldilocks yields over $70\%$ better accuracy improvement, while requiring to disclose no data about labels or label distribution.

## 2 THE LONELINESS METRIC

We consider a typical distributed ML task where a set of learning nodes $\{n^k\} \in \mathcal{N}$, equipped with local datasets $X^k$ and labels $y^k$, have to optimize the average value of loss function $L$ by choosing the weights $W$ of a parameterized learning model, i.e.,

$$\min_{W} \frac{1}{|\mathcal{N}|} \sum_{n^k \in \mathcal{N}} L(X^k, y^k, W).$$

Weights themselves can be set through any distributed learning algorithm, e.g., the classic FedAvg.

To characterize the quality of each local dataset, we start by introducing a sample-specific quantity. Specifically, we define the *loneliness* $\ell(i, k)$ of sample $x_i^k$ in dataset $X^k$ as the distance between $x_i^k$ and the closest other sample in $X^k$:

$$\ell(i, k) = \min_{x_j^k \in X^k \setminus \{x_i^k\}} \left\| x_i^k - x_j^k \right\|. \tag{1}$$

It follows from (1) that the farther away a sample is from the others, the higher its loneliness is. Samples with high loneliness might be outliers. On the contrary, samples with low loneliness are very similar to other samples—in the extreme case, repeated samples have zero loneliness.

We further extend the notion of loneliness to the dataset $X^k$ owned by node $n^k \in \mathcal{N}$, by considering the smallest sample-wise loneliness in $X^k$:

$$\hat{\ell}(X^k) = \min_{x_i^k \in X^k} \ell(i, k). \tag{2}$$

Note that the loneliness metric does not depend on the label of the points in the datasets. The relationship between sample- and dataset-wise loneliness is exemplified in Fig. 2.

To illustrate the usefulness of the loneliness metric, we conduct in the next section an experiment illustrating its impact on the performance of learning algorithms operating on small training sets.

## 3 EXPERIMENTAL ANALYSIS

**The micro-datasets.** To ascertain the effect of loneliness on the learning performance and to compare it with the traditionally used label balance, we start from the popular MNIST dataset and create a total of 90 micro-datasets, according to the following rules:

- all micro-datasets have $500$ samples, extracted from the $60\,000$ samples of the MNIST training set; every micro-dataset has a different combination of label balance and loneliness level as specified below;

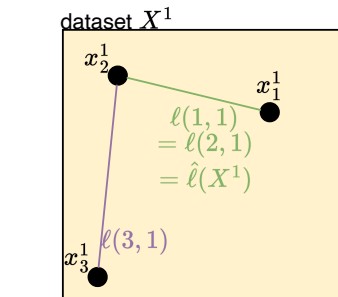 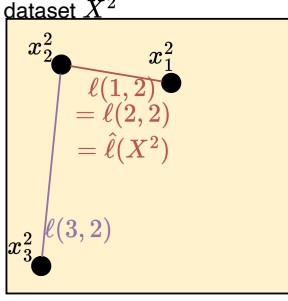

Figure 2: The relationship between distance between samples (e.g., $x_1^1$ and $x_2^1$) in a dataset, represented as a yellow box, and loneliness values. The distance from each sample to the closest sample to it corresponds to the sample-wise loneliness defined in (1); the smallest sample-wise loneliness corresponds to the dataset-wise loneliness as defined in (2).

- in each micro-dataset, one of the 10 classes is over-represented, and $\alpha \in [1, 5]$ is the unbalance factor, i.e., the ratio between the number of samples of the most and least represented classes;

- each micro-dataset $k$ has a loneliness level $\lambda(X^k)$ ranging between 1 and 10 and defined as

$$\lambda(X^k) = 1 + \left\lfloor 10 \frac{\hat{\ell}(X^k) - \min_{n^h \in \mathcal{N}} \hat{\ell}(X^h)}{\max_{n^h \in \mathcal{N}} \hat{\ell}(X^h) - \min_{n^h \in \mathcal{N}} \hat{\ell}(X^h)} \right\rfloor. \tag{3}$$

The micro-datasets so obtained reproduce those cases where there is a large number of potential learning nodes, all equipped with datasets that are (i) small and (ii) defective in different ways. In such scenarios, it is often impractical or impossible to query a large number of learning nodes. Hence, choosing them wisely is crucial, especially considering the need for dependable, sustainable learning in sensitive applications, involving heterogeneous nodes with different capabilities.

In view of the relative simplicity of the MNIST dataset, we use the LeNet5 DNN for our tests; such DNN includes three convolutional layers and two fully-connected ones. Additional details on the data and DNN models used for all tests conducted in the paper can be found in Appendix A.

**Loneliness and test accuracy.** Fig. 3 summarizes the main results of our MNIST tests. We start from the relationship between label distribution and learning performance: in Fig. 3(a) and Fig. 3(b), each line corresponds to one value of the unbalance factor $\alpha$ (in Fig. 3(a)) and loneliness level $\lambda$ (in Fig. 3(b)), and depicts the empirical cumulative density function (ECDF) of the test accuracy yielded by the corresponding micro-datasets. The value of $\alpha$ or $\lambda$ tells us on which of the ECDFs the actual accuracy is; as an example, for a loneliness level equal to 2, the accuracy is between 76% and 78% with 90% probability, and its expected value is 77.4%.

The difference between the two plots is striking: ECDFs corresponding to different values of the unbalance factor $\alpha$ in Fig. 3(a) almost always overlap and cover virtually all accuracy values on the $x$-axis. Furthermore, the mean values (i.e., the dots) are very concentrated. All values of $\alpha$ result in accuracy levels between 77% and 88% with a mean around 83%. We note that knowing the exact value of $\alpha$ does not help make that information more specific. On the other hand, the ECDFs corresponding to different values of loneliness level $\lambda$ are much more far apart, and overlap to a very limited extent. Similarly, the mean values are also far from each other (also note that we depict ten levels of loneliness, but only five values of $\alpha$). It is thus evident that loneliness offers more detailed information on where the resulting accuracy will be. Hence, it is not only a less expensive metric to compute, since it does not require labeling. It is also more useful.

Fig. 3(c) and Fig. 3(d) provide a more detailed view of this effect. Each marker in the plots corresponds to a dataset, and its position along the $x$- and $y$-axis corresponds to the value of the metric (unbalance factor $\alpha$ in the former plot, loneliness level $\lambda$ in the latter) and resulting accuracy, respectively. We note that the shaded area is almost rectangular for the unbalance factor, which confirms

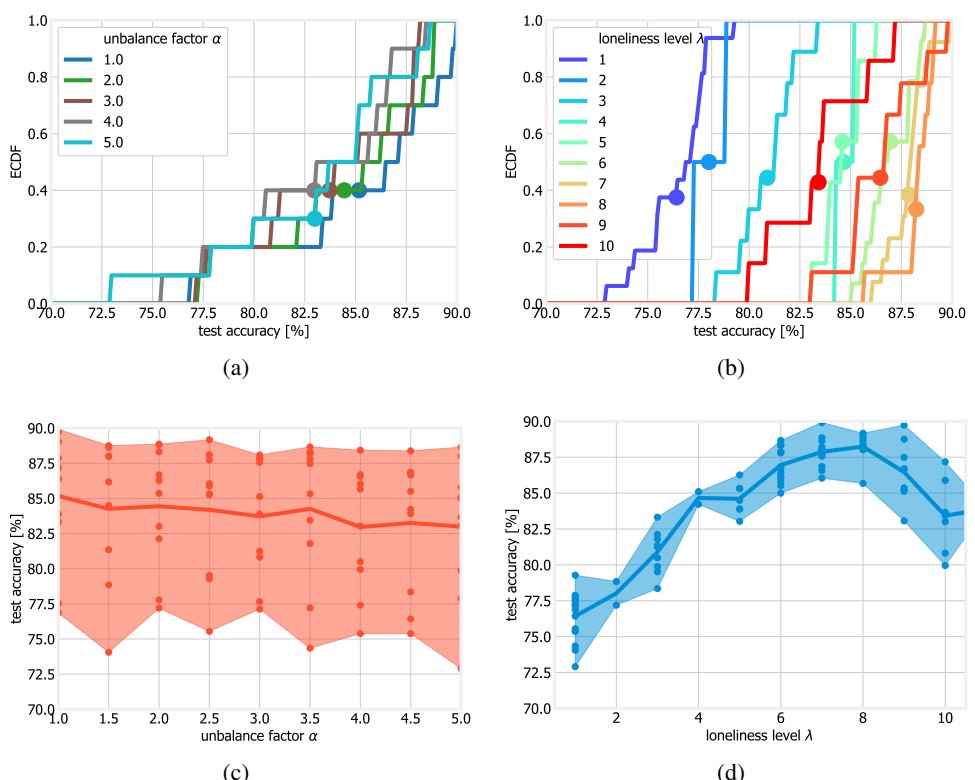

Figure 3: MNIST experiments with the LeNet DNN: loneliness is more useful than label balance as a metric to predict accuracy. Distribution of the test accuracy for different values of the unbalance factor $\alpha$ (a) and loneliness level $\lambda$ (b), with dots representing mean values; test accuracy levels achieved by micro-datasets with different unbalance factors (c) and loneliness (d).

how different values of $\alpha$ correspond to similar values of accuracy; for loneliness, instead, we observe a much more narrow, arch-like shape.

The narrowness of the shaded area in Fig. 3(d) further indicates that $\lambda$ serves as an excellent proxy for the resulting accuracy. Interestingly, the arch-like shape of the area shows that the best accuracy is achieved for *intermediate* levels of loneliness (between 5 and 7). The behavior in Fig. 3(d) makes intuitive sense if we consider that high levels of loneliness might be associated with the presence of outliers. A whole dataset composed of outliers is in fact harder to generalize from than one with a more balanced composition.

In summary, we can conclude that **loneliness has a much stronger correlation with test accuracy. Hence, it is much more useful than class balance when predicting it. Furthermore, the best accuracy is reached for intermediate levels of loneliness.**

## 4 LONELINESS THROUGH THE LENSES OF A COMPRESSION-BASED PAC-BAYES BOUND

In this section, we use a PAC-Bayes generalization bound to explain why the test accuracy of micro-datasets peaks at intermediate loneliness $\hat{\ell}$. Let $\mathcal{Z}$ be the instance space and $P_Z$ be the unknown distribution on $\mathcal{Z}$ that generates the training data $\mathbf{Z}=(Z_1, \ldots, Z_m) \in \mathcal{Z}^m$ independently. Let $\mathcal{W}$ be the hypothesis space and $P_{W|\mathbf{Z}}$ be a probabilistic learning algorithm that takes $\mathbf{Z}$ and outputs a hypothesis $W \in \mathcal{W}$. Finally, let $c : \mathcal{W} \times \mathcal{Z} \to \mathbb{R}_+$ be the loss function. Then the population loss of $w$ is defined as $L_{P_Z}(w)=\mathbb{E}_{P_Z}[c(w, Z)]$ and the training loss is defined as $L_{\mathbf{Z}}=\frac{1}{m}\sum_{i=1}^{m} c(w, Z_i)$. Given a prior distribution $Q_W$ and a posterior distribution $P_{W|\mathbf{Z}}$ over $\mathcal{W}$, McAllester (1999) states

that with probability at least $1-\delta$ under $P_{\mathbf{Z}}$,

$$\mathbb{E}_{P_{W|\mathbf{Z}}}[L_{P_Z}(W)] \leq \mathbb{E}_{P_{W|\mathbf{Z}}}[L_{\mathbf{Z}}(W)] + \sqrt{\frac{\mathrm{KL}(P_{W|\mathbf{Z}}\|Q_W) + \log(m/\delta) + 2}{2m-1}}. \qquad (4)$$

In words, the population loss can be upper-bounded by the sum of a training loss and a complexity term that quantifies, via the Kullback-Leibler (KL) distance, the penalty incurred in assuming $W \sim P_{W|\mathbf{Z}}$ when $W \sim Q_W$. Bounds on the population error similar to (4) are usually referred to as PAC-Bayes bounds.

To shed lights on the impact of the loneliness on the PAC-Bayes bound (4), we experimentally evaluate $\mathbb{E}_{P_{W|\mathbf{Z}}}[L_{\mathbf{Z}}(W)]$ and $\mathrm{KL}(P_{W|\mathbf{Z}}\|Q_W)$ for datasets with different loneliness. Specifically, following Lotfi et al. (2022), we select the universal prior $Q_W(W){=}2^{-\mathcal{K}(W)/\gamma}$ where $\mathcal{K}$ is the prefix Kolmogorov complexity of $W$ (Sunehag & Hutter, 2015) and $\gamma \leq 1$. The learning algorithm $P_{W|\mathbf{Z}}$ is chosen to be the point mass distribution on $W^\star$, which is the hypothesis obtained by training on $\mathbf{Z}$. Training is performed in two steps, according to the adaptive subspace compression algorithm introduced in Lotfi et al. (2022): (i) first, we learn a low-dimensional linear embedding of the model weights; then (ii) we quantize the embedded weights to a fixed number of levels. The training loss is computed using the 0–1 loss function. The KL term for the selected posterior and prior is upper-bounded by an expression containing the length of the shortest program needed to reproduce $W^\star$. This is computed as the number of bits required to represent the quantized model weights extracted from step (ii) of the training procedure, using an arithmetic code.

We construct 50 micro-datasets from MNIST with varying loneliness values $\hat{\ell}$ and fixed unbalance factor $\alpha{=}1$ (i.e., balanced datasets). The experimental details are provided in Appendix A. In Fig. 4, we plot the training loss and the KL term for three different values of dataset size. Each dot represents the training loss and the KL term for one dataset, respectively. We also report the linear regression line for each plot, the corresponding Pearson correlation coefficient $r$-value, and the two-sided $p$-value.

We see that for all dataset sizes $m$ considered in the figure, the training loss appears to increase as a function of the loneliness value. Intuitively, as the samples in the dataset become more dissimilar, the training process becomes slower, and the training error achieved after a fixed number of iterations, increases. On the contrary, the KL term appears to decrease monotonically with the loneliness. This suggests that the model complexity decreases. Intuitively, a dataset with a higher loneliness can be classified using a more compressible neural network.

As a result of these two opposite trends, intermediate loneliness values are preferable. Unfortunately, this cannot be demonstrated by evaluating the PAC-Bayes bound in (4) directly, since the training procedure suggested in Lotfi et al. (2022) yields vacuous results whenever the dataset size $m$ is below 2000. For $m \geq 2000$, the variation in loneliness across the generated datasets is not significant and no trends can be inferred. Indeed, this is the reason why the slope of the linear regression curve in Fig. 4 decreases when $m$ increases.

## 5 NODE AND DATASET SELECTION: GOLDILOCKS

In this section, we describe our Goldilocks node- and data-selection procedure. For the sake of simplicity, we first focus on the node selection part (i.e., we assume that all local data of selected nodes will be used), and then we will discuss data selection.

**Node selection:** The node selection problem can be stated as follows: given a set $\mathcal{N}^{\mathrm{curr}}{\subset}\mathcal{N}$ of currently-selected learning nodes, we want to identify a new node $n^*{\in}\mathcal{N}\backslash\mathcal{N}^{\mathrm{curr}}$ to add to the training process, so as to optimize the learning outcome, e.g., maximize the test accuracy. Such outcome is estimated through a *proxy metric* $\hat{\mu}(\mathcal{Q})$, taking as input a set $\mathcal{Q}$ of selected datasets, i.e., $\mathcal{Q}{=}\bigcup_q X^q$; we are also given a target value $\hat{\tau}$ for the metric $\hat{\mu}$. The metric $\hat{\mu}$ itself can correspond to any metric taking as an input a set of datasets, including the loneliness $\hat{\ell}$ defined in (2) and the unbalance factor $\alpha$. In all cases, the value of the metric of a set of datasets $\mathcal{Q}$ corresponds to the value of that metric computed over the union of all datasets therein.

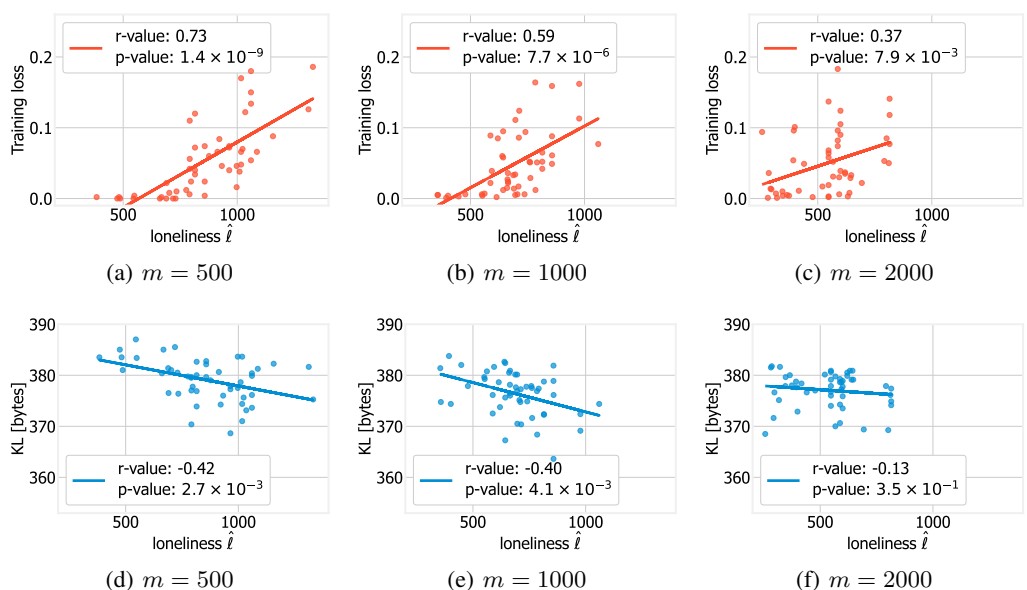

Figure 4: Training loss and KL terms in (4) for different micro-datasets generated from MNIST. The parameter $m$ indicates the size of the micro-dataset.

Given all the above, the selected node according to the Goldilocks procedure is simply the one that results in the metric $\hat{\mu}$ being closest to the target $\hat{\tau}$, i.e.,

$$n^* \leftarrow \arg\min_{n \in \mathcal{N} \setminus \mathcal{N}^{\text{curr}}} |\hat{\mu}(\mathcal{N}^{\text{curr}} \cup \{n\}) - \hat{\tau}| \,. \tag{5}$$

It is worth emphasizing that the Goldilocks procedure just outlined can be performed for arbitrary metrics, including the unbalance factor $\alpha$ introduced in Sec. 3, the loneliness level $\lambda(X^k)$ defined in (3), as well as other metrics defined in the literature, e.g., the normalized entropy used in (Bansal et al., 2023). Furthermore, Goldilocks supports arbitrary target values, i.e., it can be applied to metrics that do not need to be maximized or minimized: this is fundamental in situations like those depicted in Fig. 3(right), where the best performance is associated with intermediate loneliness values. In these cases, Goldilocks is able to select the nodes that are *just right* (hence the name of the procedure) for the task at hand.

**Data selection:** Goldilocks follows a similar approach when selecting samples $x$ within a given dataset $X^k$. Specifically, given a number $T$ of samples to select, and assuming a per-sample version $\mu(i, k)$ of the metric $\hat{\mu}$ and the corresponding target value $\tau$, Goldilocks proceeds as follows:

1. Associate with each sample $x_i \in X^k$ a score $s(i, k) = |\mu(i, k) - \tau|$;
2. Take the $T$ samples with the lowest score.

As mentioned earlier, the steps above can be applied only if the metric to use is defined for individual samples, i.e., if $\mu(i, k)$ does exist. This is the case of loneliness (as per (1)), but not, as an example, for the label balance. We can thus remark again that, by combining the loneliness metric and the Goldilocks procedure, we are able to make fine-grained, joint node and data selection decisions.

**Scope and goals of Goldilocks:** As per (5), Goldilocks selects nodes solely based on the metric $\hat{\mu}$, ignoring such factors as node connectivity, resources, and costs. All these factors need to be weighted against sheer learning performance (e.g., training accuracy) in real-world situations, as also discussed in Sec. 7. It is worth stressing that Goldilocks is not meant to give a complete, self-contained solution to the node and data selection problem in FL. Rather, it helps assessing the usefulness of (i) making node and data selection decisions jointly, and (ii) comparing different metrics, and combining them if appropriate. Such an approach can then be integrated within any of the

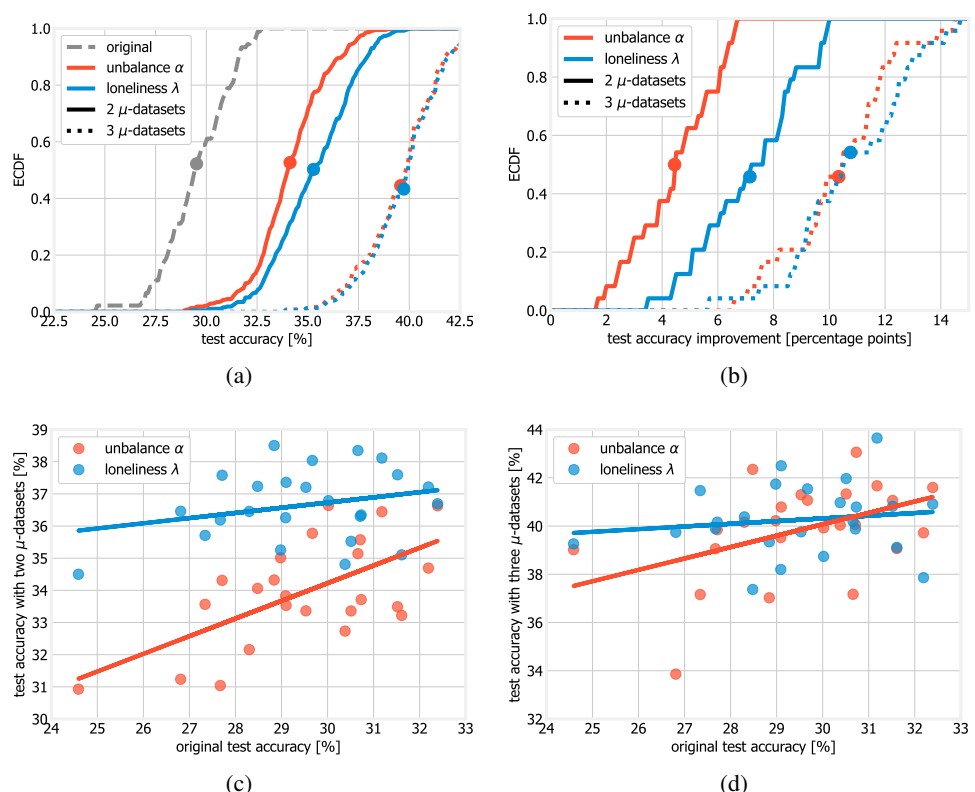

Figure 5: Goldilocks performance on the CIFAR dataset: distribution of test accuracy (a) and of accuracy improvement (b), under different metrics and for different numbers of micro-datasets; relationship between initial accuracy and accuracy improvement when adding one (c) and two (d) micro-datasets.

alternative approaches discussed in Sec. 7, leaving the remaining parts thereof (dealing, for example, with connectivity) in place.

# 6 PERFORMANCE EVALUATION

In the experiment reported below, we verify the usefulness of the Goldilocks procedure in the context of node selection, especially when loneliness is chosen as metric. To this end, we perform a set of experiments using the CIFAR10 dataset (Krizhevsky et al., 2009) and the MobileNetV2 DNN (Dong et al., 2020). Specifically, (i) We generate from the CIFAR10 training set $100$ micro-datasets with $500$ samples each, following the same procedure as in Sec. 3; every micro-dataset has different loneliness and label unbalance factor. (ii) For each micro-dataset, we use the Goldilocks procedure described in Sec. 5, in combination with either the label unbalance factor $\alpha$ or the loneliness level $\lambda$, to select one or two additional datasets. (iii) We train the model using each combination of datasets, in a centralized manner, for $50$ epochs, tracking the resulting test accuracy. For further details about the experiments, see Appendix A.

Fig. 5(a) shows the distribution of accuracy for one (dashed line), two (solid lines), and three (dotted lines) micro-datasets. The color of the lines reflects the metric employed to select the additional data: red for the label unbalance $\alpha$ (for which a target of $\hat{\tau}=1$ is set), and blue for the loneliness level $\lambda$ (for which we set, based upon Fig. 3, $\hat{\tau}=8$). As expected, adding more data results in better accuracy. More interestingly, using loneliness *in lieu* of label unbalance results in a significantly better accuracy, for the same quantity of data.

Fig. 5(b), which summarizes the accuracy improvement, offers a more detailed view. We can observe that adding one micro-dataset to the training yields an average accuracy improvement of $4\%$ when

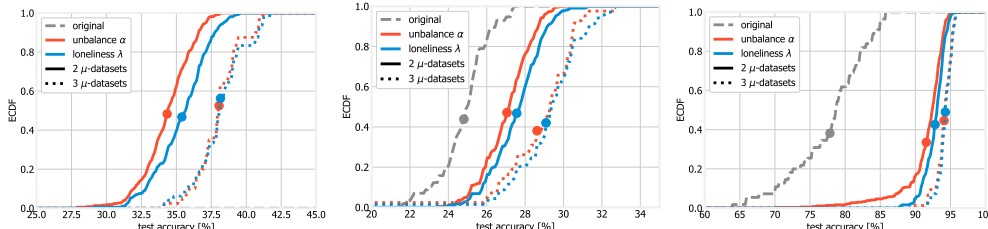

Figure 6: Distribution of the test accuracy achieved for different numbers of micro-datasets and metrics in a FL scenario using the CIFAR dataset (left), a centralized scenario using the CINIC10 dataset (center), and a centralized scenario using the GTSRB dataset (right).

the micro-dataset is chosen considering the label unbalance factor $\alpha$, and over $7\%$ when loneliness level $\lambda$ is accounted for. In a situation where the datasets are not labeled and labeling is expensive, a Goldilocks procedure based on loneliness has the additional advantage that only the selected datasets will need to be labeled. On the contrary, a procedure based on the class imbalance factor would require one to label all datasets. This is especially advantageous in training scenarios where only few datasets can be used due to complexity constraints.

Next, in Fig. 5(c) and Fig. 5(d), we look in more detail at *when* extra data result in the largest accuracy improvement. Each marker in the plots corresponds to a micro-dataset, and its position along the $x$- and $y$-axes corresponds to the accuracy obtained using that micro-dataset alone and combining that dataset with additional one(s) using the Goldilocks procedure, respectively; the color of the marker corresponds to the metric employed. We can observe that datasets with lower accuracy tend to benefit the most from extra data. It is also interesting to observe how the improvement yielded by loneliness over label unbalance factor is larger when we have to select two micro-datasets than three. This confirms that loneliness is especially useful when the quantity of data (and the number of nodes) that can be selected is small.

## 6.1 FL AND ADDITIONAL DATASETS

To better assess how general our results are, we extend our performance evaluation to:

- A FL scenario still using CIFAR, where each micro-dataset belongs to a different learning node and learning nodes cooperate via the FedAvg algorithm;
- A centralized scenario using the CINIC10 dataset (Darlow et al., 2018), developed as a more diverse, drop-in alternative to CIFAR;
- A centralized scenario using the GTSRB dataset (Stallkamp et al., 2011), including real-world pictures of road signs.

In FL, nodes cooperate through the FedAvg algorithm as implemented by the `flwr` library. Additional details on the datasets and implementation can be found in Appendix A.

The test accuracy achieved in the aforementioned cases is summarized in Fig. 6. Consistently with Fig. 5(a), more data always result in better accuracy; also, using loneliness as a metric yields consistently a better accuracy. In agreement with Fig. 5(c) and Fig. 5(d), the effect is more significant when choosing two micro-datasets than when choosing three. We refer the reader to Appendix B for more details on the FL, CINIC10, and GTSRB experiments and results.

## 7 RELATED WORK

The problem of node selection in distributed learning and of dataset selection in conventional learning is well investigated. We provide below some relevant works while highlighting the difference with the present contribution.

**Node selection:** The problem of node selection in cross-device FL is particularly relevant in the presence of data heterogeneity, which causes FedAvg to suffer from client drift (Karimireddy et al.,

2020). Data heterogeneity, combined with sporadic client participation, causes also a participation gap, on top of the usual generalization gap in statistical learning (Yuan et al., 2022). To estimate this participation gap, Yuan et al. (2022) suggest to let each client share some held-out data with the aggregator. The availability of held-out data at the aggregator enables the implementation of sophisticated client-selection strategies, which, as shown in Singhal et al. (2024), improve communication efficiency. However, sharing such data seems to defy one of the main purposes of FL, which is to maintain data privacy at the clients. Cho et al. (2022) propose a different approach, in which client selection at the aggregator is performed on the basis of the local training loss computed at each client, which does not require sharing held-out data. Note that, differently from our much-simpler loneliness based approach, both solutions require one to label all datasets, including the ones that will not be used.

**Data selection:** This is often referred to as data pruning in the literature (Sachdeva & McAuley, 2023), and involves pruning low-quality data, typically during the training process. An extensive review of data pruning methods can be found in Guo et al. (2022). Among the existing solutions, Ghorbani & Zou (2019) uses the Shapely-value of a subset of data to estimate their utility. Toneva et al. (2019) remove from the training set unforgettable examples, i.e., examples whose predicted label is correct and does not change over the training process. Paul et al. (2021), instead, retains "hard" examples, i.e., examples that have large $\ell_2$-norm scores on trained models. A different approach is used in CRAIG (Mirzasoleiman et al., 2020), which selects a subset of the training data that closely approximates the full gradient. Unlike these approaches, the much simpler loneliness-based method proposed in this paper does not require knowledge of the sample labels and/or of the learning algorithm.

## 8 CONCLUSION

We have considered the problems of node and data selection in cooperative learning scenarios where only a small number of small datasets can be used. In this scenario, it is especially advantageous to select the datasets (and, hence, the nodes) *without* using label information, for both efficiency and privacy reasons. To this end, we proposed a metric called *loneliness*, which, using unlabelled data, computes the distance between samples of the same dataset. We integrated loneliness with a node- and data-selection strategy called Goldilocks, and found – across multiple datasets and in both centralized and federated schemes – that using loneliness consistently results in higher improvement of training accuracy, by over 70%, in spite of requiring no label information.

### REPRODUCIBILITY STATEMENT

All datasets used for all experiments are publicly available. The complete code (Python and Jupyter notebooks) needed to obtain all the results of this paper (including the Appendix) is available at the anonymized repository
`https://anonymous.4open.science/r/Choose-Before-You-Label-5701/`.

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

## A    EXPERIMENT DETAILS

**Datasets, architectures, meta-parameters:**    For our experiments, we consider the following datasets:

1. MNIST (Kayed et al., 2020), including $70\,000$ $28 \times 28$ black-and-white images of hand-written digits. The dataset is divided into a training set of $60\,000$ images and a testing set of $10\,000$ images, and comprises 10 classes, one per digit.

2. CIFAR (Krizhevsky et al., 2009), including $60\,000$ $32 \times 32$ color images of different objects and animals (airplanes, cars, birds, cats, deer, dogs, frogs, horses, ships, and trucks). The dataset is divided into a training set of $50\,000$ images and a testing set of $10\,000$ images, and comprises 10 classes, one per object.

3. CINIC (Darlow et al., 2018), including $270\,000$ $32 \times 32$ color images: the $60\,000$ ones from CIFAR, plus $210\,000$ coming from ImageNet. It includes the same classes as CIFAR and is divided into train, test, and validation splits, each including $90\,000$ images. It is meant as a drop-in, more challenging replacement to CIFAR.

4. GTSRB (Stallkamp et al., 2011) (German Traffic Sign Recognition Benchmark), including over $38\,000$ color, real-world images of road signs, belonging to $40$ classes. Different images may have different size, hence, we need to resize them to $32 \times 32$; the dataset is divided into a training set of $26\,640$ images and a testing set of $12\,630$ images.

For MNIST, we use the LeNet5 convolutional network (Kayed et al., 2020), including three convolutional layers and two fully-connected ones. For all other datasets, we use MobileNetV2 (Dong et al., 2020). In all scenarios, we train for 50 epochs, and consider the one yielding the best performance. Stochastic gradient descent is used as an optimizer, with a learning rate of $10^{-3}$. All experiments are performed with PyTorch, and training employs the `lightning` library.

**Evaluation of the PAC-Bayes bound in** (4)**:**    We use the LeNet model (LeCun et al., 1998) as the base model. Following (Lotfi et al., 2022, Sections 4.1 & 4.2), we train a compressed model (compressed size = 1000) for 1000 epochs using the Adam optimizer with a learning rate of 0.001. Then a quantized model (quantization levels = 7) is trained for 30 epochs using the Adam optimizer with a learning rate of 0.0001. The cross-entropy loss is used during training, and the 0–1 loss is used to compute the training loss in (4). The KL term in (4) is approximated as follows. Let $W^*$ be the trained model obtained after the compression and quantization. Then the posterior is set as $P_{W|\mathbf{z}} = \mathbb{I}_{W=W^*}$, where $\mathbb{I}$ is the indicator function. The KL term is then bounded as

$$\mathrm{KL}(P_{W|\mathbf{z}}\|Q_W) = \mathrm{KL}(\mathbb{I}_{W=W^*}\|2^{-\mathcal{K}(W)/\gamma}) = \log\left(\frac{1}{2^{-\mathcal{K}(W^*)/\gamma}}\right)$$

$$\leq \mathcal{K}(W^*)\log 2 \overset{(\dagger)}{\leq} \rho(W^*)\log 2 + 2\log\rho(W^*), \tag{6}$$

where $\rho(W^*)$ is the number of bits required to represent the weights of $W^*$ and to obtain $(\dagger)$ we proceeded as in (Cover & Joy, 2006, Theorem 14.2.3). We use arithmetic coding to compute $\rho(W^*)$.

## B    FULL RESULTS FOR THE FL, CINIC, AND GTSRB EXPERIMENTS

### B.1    FULL RESULTS FOR THE FL EXPERIMENTS

The full results of the FL experiments, using micro-datasets drawn from the CIFAR10 dataset, are reported in Fig. 7, and are consistent with those of Fig. 5 in the main paper.

### B.2    FULL RESULTS FOR THE CINIC AND GTSRB EXPERIMENTS

The full results of the CINIC and GTSRB experiments, are reported in Fig. 8 and Fig. 9, and are consistent with those of Fig. 5 in the main paper.

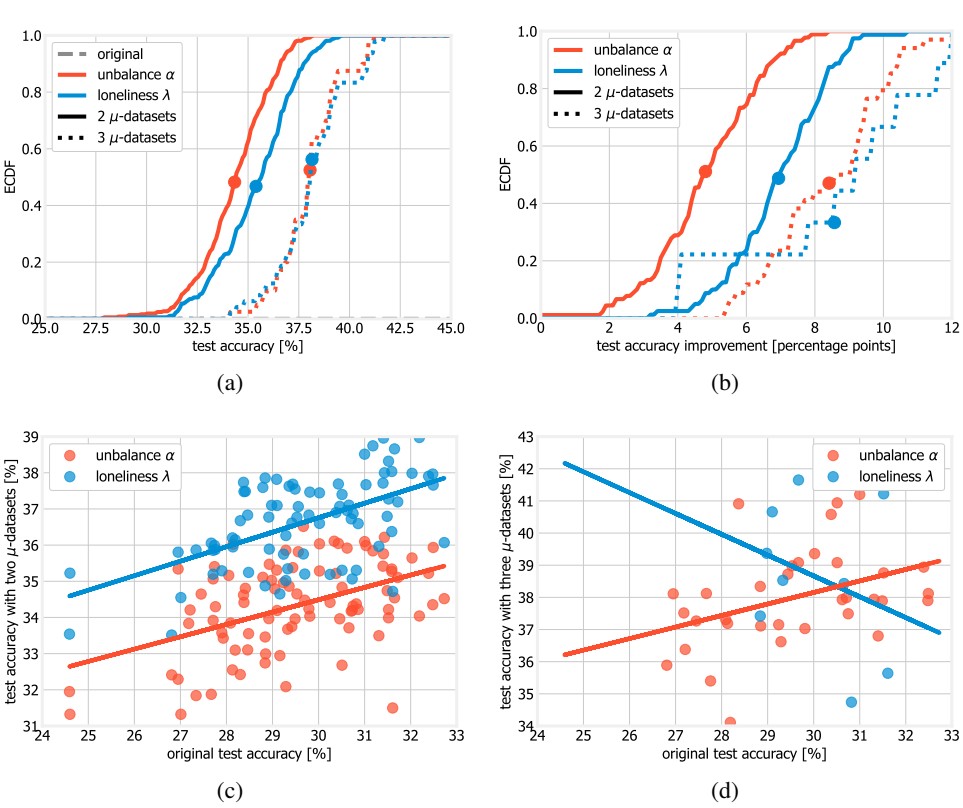

Figure 7: Goldilocks performance on the CIFAR dataset for a FL scenario: distribution of test accuracy (a) and of accuracy improvement (b) under different metrics and for different numbers of micro-datasets; relationship between initial accuracy and accuracy improvement when adding one (c) and two (d) micro-datasets.

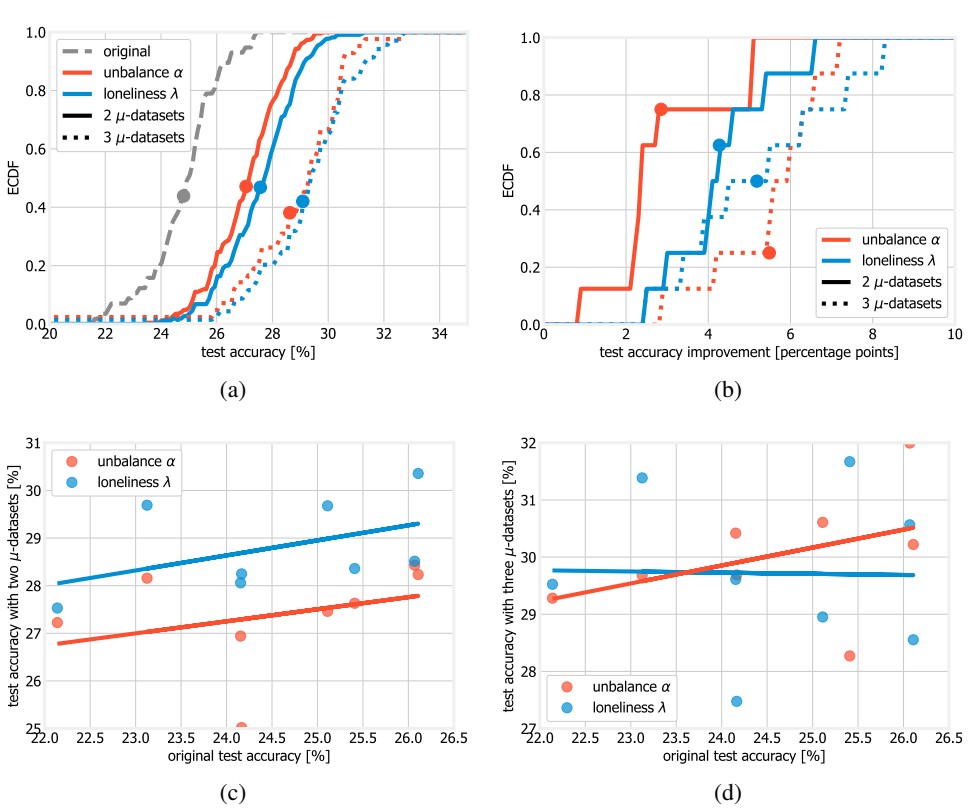

Figure 8: Goldilocks performance on the CINIC dataset: distribution of test accuracy (a) and of accuracy improvement (b) under different metrics and for different numbers of micro-datasets; relationship between initial accuracy and accuracy improvement when adding one (c) and two (d) micro-datasets.

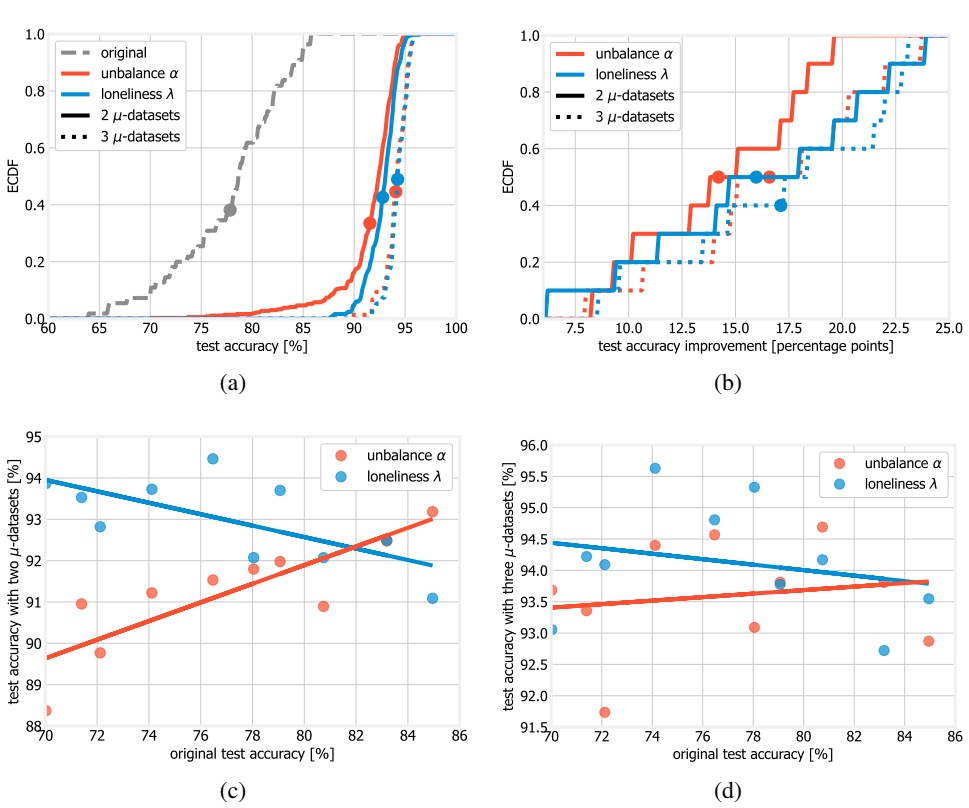

Figure 9: Goldilocks performance on the GTSRB dataset: distribution of test accuracy (a) and of accuracy improvement (b) under different metrics and for different numbers of micro-datasets; relationship between initial accuracy and accuracy improvement when adding one (c) and two (d) micro-datasets.

