# OpenReview forum: "Choose Before You Label: Efficient Node and Data Selection in Distributed Learning"
_ICLR.cc/2025/Conference — ICLR 2025 Conference Withdrawn Submission_

### Official Review · Reviewer_Fjga · 2024-10-30

**Soundness:** 2
**Presentation:** 3
**Contribution:** 1
**Rating:** 3
**Confidence:** 4

**Summary:**

The paper studies a problem of node and data selection in distributed learning and propose a metric, called loneliness, to measure the quality of data samples. The loneliness of a data sample quantifies its distance to the nearest neighboring sample within the same dataset. Extending this, the loneliness of a dataset is defined as the minimum loneliness across all sample pairs within it. Experiments are conducted to verify loneliness correlates with test accuracy, training loss, and KL divergence.

**Strengths:**

S1. The paper proposes an interesting metric called loneliness to measure the value of unlabeled training samples.
S2. The paper is well written, with experiments whose details and evaluations are clearly described.

**Weaknesses:**

W1. Pairwise distances are widely studied, particularly in the context of clustering tasks. K nearest neighbors are commonly used in outlier detection. The authors can further discuss the comparison of loneliness to existing distance-based measures.
W2. Experiments in the paper mainly focus on validating the correlation of loneliness with key metrics, such as test accuracy, training loss, and KL divergence. However, it lacks comparisons with state-of-the-art methods for node and data selection.
W3. While the authors find that intermediate levels of loneliness value can achieve the best performance, the paper would benefit from a deeper exploration of how to determine the optimal loneliness value in node or data selection.
W4. In a federated learning scenario, where privacy is essential, it is unclear how a central server can collect loneliness information from clients.

**Questions:**

pls see weaknesses

---

### Official Review · Reviewer_YT8e · 2024-10-31

**Soundness:** 2
**Presentation:** 2
**Contribution:** 2
**Rating:** 3
**Confidence:** 3

**Summary:**

The paper presents "loneliness," a metric for selecting nodes and data in distributed learning that prioritizes privacy and resource efficiency. Unlike traditional balanced label approaches, loneliness measures distances between unlabelled samples to assess dataset quality. The proposed Goldilocks method applies this metric to optimize node and data selection, improving accuracy and reducing resource demands across centralized and federated learning scenarios.

**Strengths:**

- The concept of loneliness as a dataset quality metric is innovative, particularly in the context of unlabelled, privacy-preserving data selection.

- By focusing on unlabelled data, the approach reduces labeling costs and data transmission requirements, which is valuable for distributed and federated learning.

- The approach can generalize to various machine learning contexts, including centralized and federated frameworks, demonstrating flexibility.

**Weaknesses:**

- The proposed metric loneliness depends on the embedding space distance between the features of two samples, which is intuitive. However, due to privacy concerns, direct access to the samples' information—including features and labels—is restricted.

- The defined loneliness level for dataset $\lambda(X^k)$ (i.e., Eq. (3)) lacks a clear intuitive explanation and rationale. For example, why not use the KNN embedding distance to measure the loneliness for a given sample?

- The calculation of loneliness introduces significant computational overhead, which may become a non-negligible concern in resource-constrained federated learning settings. Therefore, a detailed analysis of the computation cost and time required for loneliness calculation is essential.

**Questions:**

- What is the rationale behind for defining the loneliness level $\lambda(X^k)$, as shown in Eq. (3)?

- Why not directly use the KNN distance to calculate simple-wise loneliness, $\ell(i, k)$? From my understanding, $\ell(i, k)$ could be interpreted as a 1-NN distance. In general, using a larger K could provide more stability than relying on just a single neighbor.

- The dataset-level loneliness is defined as the smallest sample-wise loneliness, which may be less meaningful in practice. In a relatively large dataset, the majority of samples tend to have similar feature distributions, making the minimum value an unreliable measure. Using the average loneliness value would likely provide a more reasonable and representative metric.

- What is the exact definition of metric $\hat{\mu}$? The explanations provided in line 321 is unclear.

---

### Official Review · Reviewer_C2FL · 2024-10-31

**Soundness:** 1
**Presentation:** 2
**Contribution:** 1
**Rating:** 3
**Confidence:** 4

**Summary:**

The paper proposes an approach to select nodes and datapoints in a federated learning scenario based upon nearest-neighbor-distance. They use the term loneliness for this nearest-neighbor distance and define loneliness of a dataset as the minimum nearest-neighbor-distance of its datapoints. Supposed we apriori know an ideal value of this loneliness, we can then select nodes with a loneliness value close to that optimum, as well as datapoints within a local dataset with a nearest-neighbor-distance close to that optimum. The authors term this selection strategy Goldilocks.

The paper motivates this procedure by arguing that sharing class imbalances of local datasets, which is typically used to select nodes, would infringe privacy, whereas their loneliness metric would not. Furthermore, they argue that selecting nodes and datapoints with an optimal loneliness value improves federated training.

The claims are partially substantiated by an empirical evaluation that shows that selecting nodes and datasets in this manner improves centralized and federated training.

**Strengths:**

- The paper tackles the relevant problem of node and dataset selection in federated learning.

**Weaknesses:**

- The technical contribution is limited, since loneliness is simply nearest-neighbor distance and Goldilocks procedure's reliance on prior knowledge of an "ideal" metric value is a major flaw, as this is the very information we seek in practice.
- The paper assumes that sharing label fractions infringes on privacy whereas sharing nearest-neighbor distances does not. This claim requires substantiation, since label fractions typically pose minimal privacy risk. This could be achieved by a theoretical or empirical analysis of the difference in privacy risks.
- The use of the minimum loneliness value per dataset is problematic due to the impact of duplicates. Additionally, norms like $L_2$ lose discriminatory power in high dimensions, so for most deep learning inputs, the effectiveness of the metric is questionable. Please justify those design decisions.
​- Metrics like local outlier factor and ECS are entirely omitted, weakening the claim that loneliness is uniquely valuable.
- Ignoring datasets with low loneliness may exclude samples that are critical for representing the overall distribution, posing a risk of overfitting to selected "easy" subsets.

**Questions:**

- Why is sharing the label fractions of nodes infringing privacy and why does loneliness as a metric reveal less about the data than label fractions?
- What if a local dataset is not selected for its loneliness score but it would be crucial for the performance on the overall distribution to train on it? Don't we run the risk of overfitting on a subset of "easy" datasets?
- Which norm is used in Eq. 1? Isn't it problematic to use a simple norm, such as $L_2$, for high-dimensional inputs, since norms become similar in high dimensions? That is, for large image data, it is likely that all images are equally lonely. (do they perform a good evaluation of their metric?)
- Why use the minimum over the entire dataset? This means I can set the loneliness of any dataset to 0 by simply cloning one datapoint.
- Why is it a good thing that loneliness does not depend on a label?
- Why is loneliness better than things like local outlier factor (cf. [1,2,3] f, or metrics like QI^2 [4] or ECS [5]?
- MNIST is probably unsuitable for a realistic evaluation of loneliness, since nearest neighbors suffice to explain the label in that case (k-nearest neighbour achieves SOTA performance on this dataset). It is likely that loneliness performs worse if the structures in input space are more complex.
- Why would you measure loneliness in input space? The goal of representation learning is to transform the potentially complex input space into a more benign feature space. Thus, high loneliness in input space might be meaningless if we can train a neural network to map into a feature space with more benign properties (cf. [6]).
- In your experiments, you use a Dirichlet distribution to simulate label heterogeneity with $\alpha\in [1,5]$, which corresponds to fairly mild heterogeneity. It would be more interesting to see results for $\alpha << 1$, e.g., the typically used $\alpha = 0.01$ in most heterogeneous FL papers.
- There is no systematic evaluation of loneliness as a metric, i.e., analyzing how the metric behaves for different datasets, how it changes with noise added to features, or other perturbations that might affect federated learning.
- The theoretical result you rely on basically says that the more outliers a dataset has, the harder it is to fit (since the KL divergence is $-\log(Q_W(W^*))$ which is simply model complexity at the optimum). This is fairly unsurprising. Also, it seems that for larger datasets, both KL divergence and loneliness become less useful as predictors of model performance.
- The Goldilocks procedure assumes that we already know an ideal value of the metric we evaluate. This, however, is the hard part. Selecting nodes or datapoints with as close an ideal value as possible is trivial. So without a method to infer upon the optimal value $\tau$, the approach only works if we apriori know the optimal value of the metric. Why is this a realistic assumption? How sensitive is the approach to getting that value right? And how can we estimate it in practice in a realistic application scenario?
- If small datasets are of concern, the paper should compare node and dataset selection with baselines for this scenario, such as FedDC [7].

**References:**

[1] Alghushairy, Omar, et al. "A review of local outlier factor algorithms for outlier detection in big data streams." Big Data and Cognitive Computing 5.1 (2020): 1.

[2] Wang, Gang, and Yufei Chen. "Robust feature matching using guided local outlier factor." Pattern Recognition 117 (2021): 107986.

[3] Xu, He, et al. "Outlier detection algorithm based on k-nearest neighbors-local outlier factor." Journal of Algorithms & Computational Technology 16 (2022): 17483026221078111.

[4] Geerkens, Simon, et al. "qi 2: an interactive tool for data quality assurance." AI and Ethics 4.1 (2024): 141-149.

[5] Sieberichs, Christian, et al. "ECS: an interactive tool for data quality assurance." AI and Ethics 4.1 (2024): 131-139.

[6] Petzka, Henning, et al. "Relative flatness and generalization." Advances in neural information processing systems 34 (2021): 18420-18432.

[7] Kamp, Michael, Jonas Fischer, and Jilles Vreeken. "Federated Learning from Small Datasets." Eleventh International Conference on Learning Representations. OpenReview. net, 2023.

---

### Official Review · Reviewer_eou8 · 2024-10-31

**Soundness:** 1
**Presentation:** 2
**Contribution:** 1
**Rating:** 1
**Confidence:** 4

**Summary:**

This paper defines a metric to evaluate and improve node/data selection in distributed learning.

**Strengths:**

The paper is easy to follow.

**Weaknesses:**

I am afraid that this paper does not meet the publication standards of ICLR.

- A key motivation in distributed learning is privacy preservation. However, authors' proposed metric requires sharing of features, which is NOT acceptable in distributed learning.

- The loneliness metric is not a novel concept. It has been looked at in clustering/unsupervised learning community decades ago. To the reviewer's point of view, there is no significant contribution in this submission.

- The proof does not associate well with the analysis. Actually, the reviewer cannot even find the proof in the submission, either in main text of appendix.

- The experiments are far from being sufficient.
  -- No existing data valuation/node selection methods are compared. --The setting is too simple in terms of datasets and models. -- No ablation study is given. -- It is inappropriate to call LeNet or MNIST as SOTA models/datasets.

**Questions:**

See weaknesses

---

### Note · Authors · 2024-11-13

I have read and agree with the venue's withdrawal policy on behalf of myself and my co-authors.